# Longitudinally Extensive Transverse Myelitis Associated with Cytomegalovirus Infection in an Immunocompetent Patient

Raul Montalvo [1,2] and María-Fernanda Sánchez Vidal [2,*]

1   Infectology Department, Universidad Cesar Vallejo, Trujillo 13001, Peru
2   Departamento Neuroinfectología, Instituto Nacional de Neurología y Neurocirugía,
    Mexico City 14269, Mexico
*   Correspondence: mariafernandasanchezvidal@gmail.com

**Abstract:** Longitudinally extensive transverse myelitis (LETM) associated with cytomegalovirus infection is extremely rare and is, mainly observed in immunocompetent people. A 55-year-old woman with no evidence of immunosuppression was admitted with paresthesia in the lower limbs for 15 days, difficulty walking, fecal incontinence, and acute urinary retention. Magnetic resonance imaging (MRI) in the T2 sequence showed signs of hyperintensity in the cervical and thoracic cord. The serological study showed IgM antibodies to cytomegalovirus, and the result of the FilmArray meningitis/encephalitis panel showed the presence of cytomegalovirus. She received treatment with methylprednisolone and ganciclovir with a favorable outcome. This case highlights the importance of investigating treatable causes in patients with longitudinally extensive transverse myelitis with acute presentation.

**Keywords:** transverse myelitis; cytomegalovirus; immunocompetent; infection





## 1. Introduction

Transverse myelitis (TM) is the inflammation of the spinal cord which causes bilateral sensorimotor dysfunction. The characteristics for diagnosis are the presence of a sensory level, an inflammatory profile of the cerebrospinal fluid (CSF) and disease progression between 4 and 21 days [1,2]. The incidence is one to four cases per million people, and it is classified according to the clinical severity and the radiological extension of the spinal cord injury: partial when its condition is mild or asymmetric; complete when it produces motor, sensory, and autonomic paralysis below the level of the lesion that affects one or two vertebral segments and longitudinally extensive transverse myelitis (LETM) when the lesion extends from three to more vertebrae [3,4]. For the diagnosis, it is important to rule out other causes of spinal cord compression such as malignancies, and the etiology of LETM can be demyelinating, autoimmune, infectious, or paraneoplastic [5,6].

Infectious agents are the first cause of TM that generally affect immunosuppressed people. One of these pathogens is cytomegalovirus, an opportunistic agent that has rarely been described as an infectious cause of transverse myelitis in immunocompetent people [7–10]. We report the case of a woman who developed LETM secondary to CMV infection.

## 2. Case Presentation

*Case*

A previously healthy 55-year-old woman was admitted to the Hospital Daniel Alcides Carrión, located in Huancayo, Perú, in the emergency department after a 15-day history of illness that began with back pain, progressive lower limb weakness, difficulty walking, and the inability to stand.

On the day of admission, there was evidence of an acute urinary retention, requiring urinary catheterization. The patient did not have symptoms of respiratory infection, diarrhea, trauma, or medication use before the onset of weakness.

The physical examination on admission revealed the patient was afebrile, had a heart rate of 92 per minute, a respiratory rate of 19 per minute, blood pressure of 118/74 mmHg, and oxygen saturation of 95%.

There were not any abnormal findings in the neck, thorax, and abdomen. Lymphadenopathy was not observed.

Neurological examination showed Glasgow Coma Scale 15, a normal eye assessment with no signs of optic neuritis or uveitis, higher brain function, and her cranial nerves were intact. The patient presented hypotonia of the lower limbs, with bilateral weakness in approximately 1/5 strength, increased tendon reflexes, and a positive Babinski sign in both legs. Hypotonia and anesthesia were found to be below the level of the dermatome of the third dorsal; no signs of meningeal irritation were observed.

## 3. Results

### 3.1. Laboratory

The results of the laboratory tests showed leukocytes $10.32 \times 10^3$/uL, neutrophils 77.9%, lymphocytes 14.8%; the C-reactive protein and erythrocyte sedimentation rate were within normal values. Serological tests for HTLV, HIV1-2, hepatitis B, hepatitis C, herpes simplex virus (HSV)-1-2, syphilis (VDRL rapid test was performed, which is a non-treponemal test), and Epstein–Barr were all negative, as were tests for antinuclear antibodies against double-stranded DNA and cytoplasmic, perinuclear, and antiphospholipid antibodies. IgM antibodies to herpes I and II, Epstein–Barr were non-reactive, but serological IgM antibody values for CMV were 37 (normal value < 22) and IgG was > 250 (normal value < 6). Serological CMV tests in blood, anti-MOG, and anti-AQP4 antibody were not performed. The elevation of liver enzymes was not observed. Testing for West Nile virus/arboviruses was not carried out because of the absence of relevant data such as travel or residence in endemic areas. Additionally, the Lyme test was not performed due to the low prevalence of the disease in Peru. A retrospective serum survey in Peru performed on 216 individuals found 4 (2%) to be positive for B.b. antibodies via ELISA. The presence of antibodies against B.b. was found in 10% of 232 otherwise healthy subjects in northern Peru using ELISA [11].

### 3.2. Image Studies

The chest and brain tomography did not display any abnormalities. The spinal MRI performed on the second day of hospital admission showed irregular intramedullary signal enhancement on the T2 sequence with diffuse lesions ranging from the sixth cervical vertebra to the seventh dorsal vertebral (Figure 1). The T1 sequence showed the spinal cord with multiple hypointense lesions centrally located at the cervical–dorsal level (Figure 2), while the transversal image T2-weighted image of the spinal cord showed hyperintense lesions in the white matter and in the posterior portion of the spinal cord (Figure 3).

### 3.3. Cerebrospinal Fluid

The analysis of cerebrospinal fluid (CSF) showed eight predominantly mononuclear leukocytes, protein 52 mg/dL, and glucose 63 mg/dL. The bacterial, mycobacterial, and fungal cultures of CSF remained negative, as did the Gram stain, India ink preparation, and acid-fast bacilli stains of CSF. The result of the FilmArray meningitis/encephalitis panel (CMV, enterovirus, VHS-1, VHS-2, HHV-6, VZV, bacterial, and yeast) (MEP, BioFire Diagnostics/Biomerieux, Salt Lake City, UT, USA) detected the presence of cytomegalovirus; the rest were non-reactive.

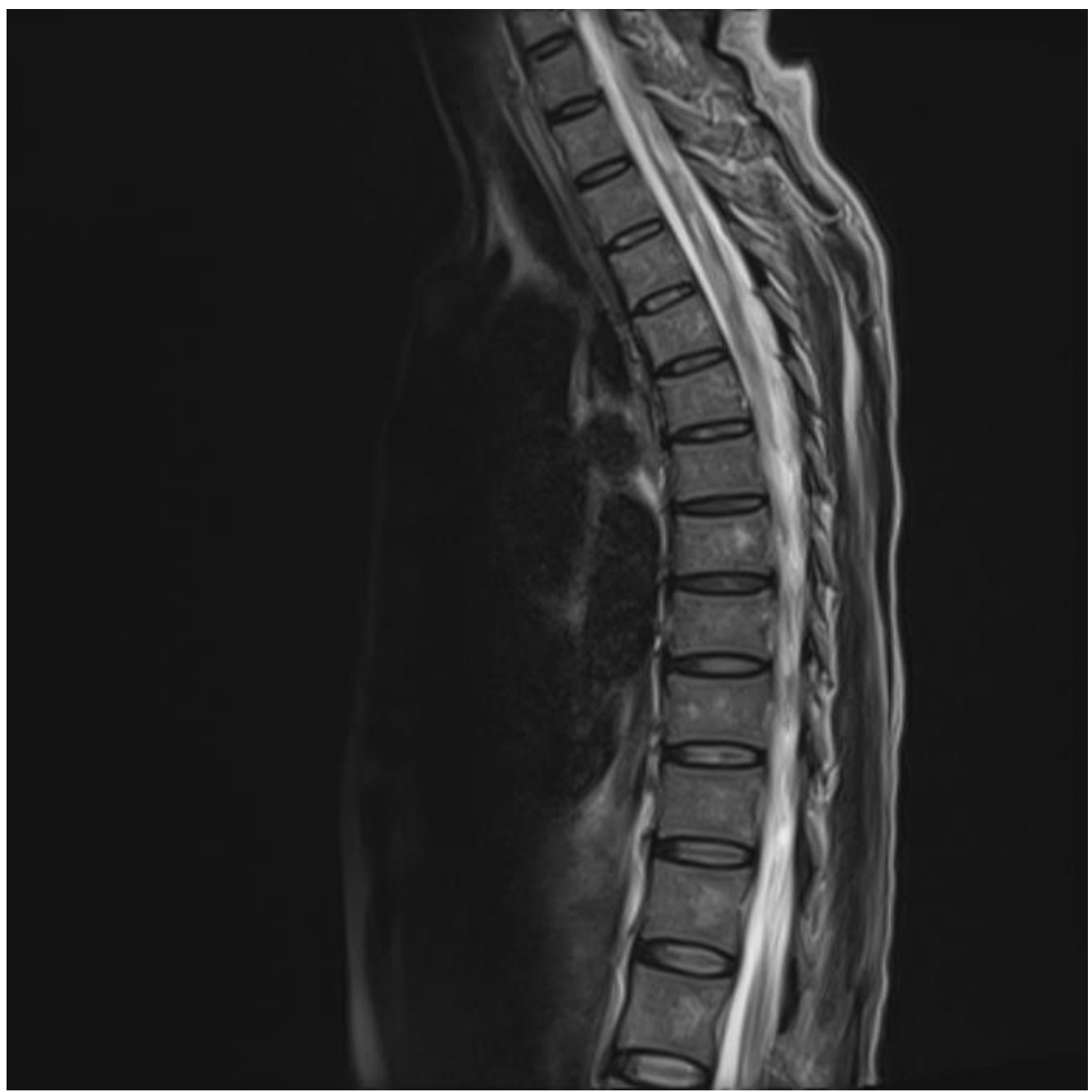

**Figure 1.** T2 image showing hyperintensity of the central portion of the cervical and thoracic cord.

With these findings, the diagnosis of LETM was made in a patient with no evidence of immunosuppression. Ganciclovir treatment (250 mg, intravenously every 12 h for 14 days) was started, as well as methylprednisolone (1g, intravenously every 24 h for 5 days). The patient presented clinical improvement and was progressed to maintenance doses with 900 mg of valganciclovir orally every 24 h for 14 days and additional daily doses of 50 mg of prednisone with a gradual reduction. During the outpatient controls, improvement was observed: she was able to stand and walk on her own.

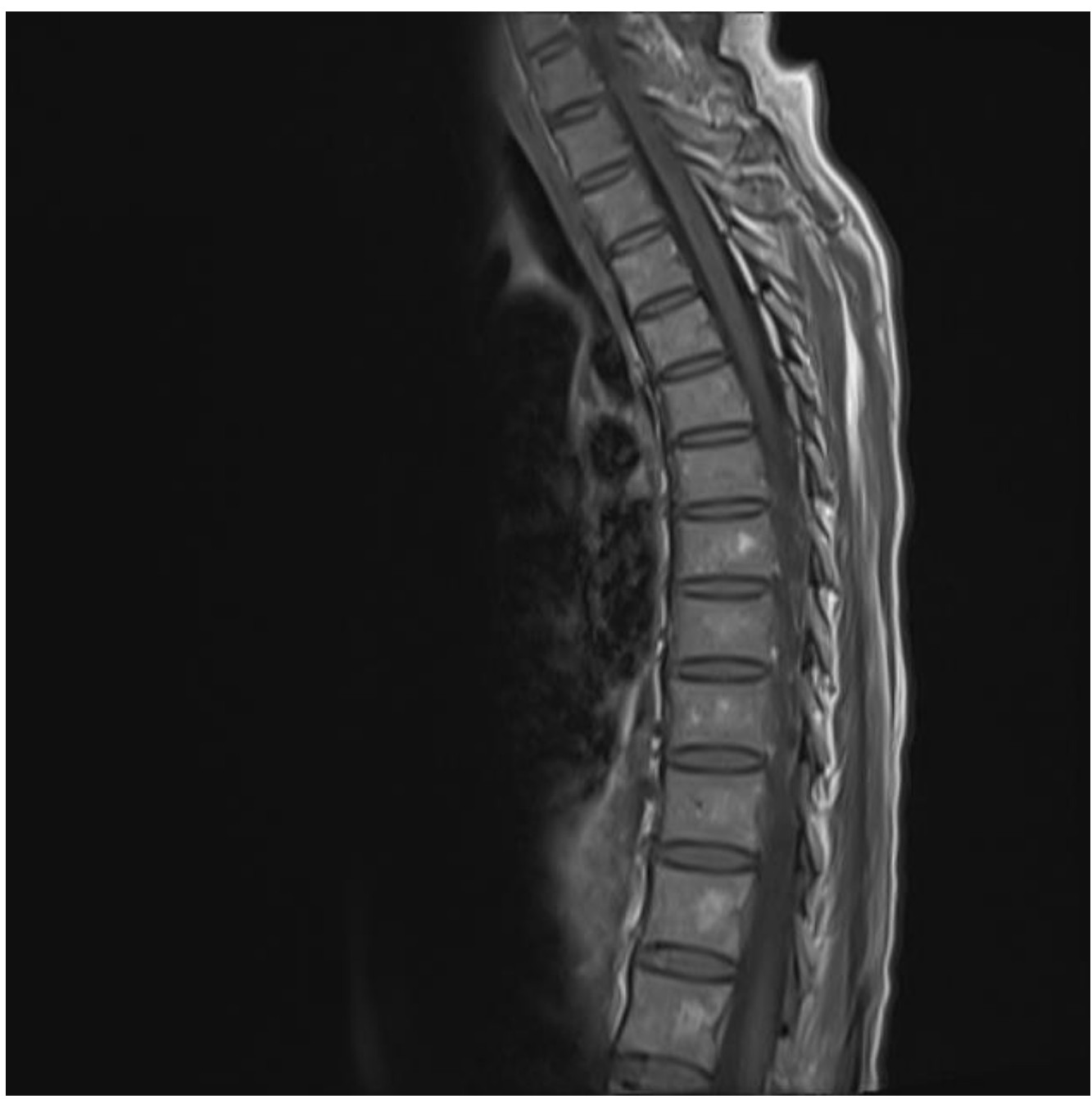

**Figure 2.** T1 image showing hypodense lesion in the cervical and thoracic cord.

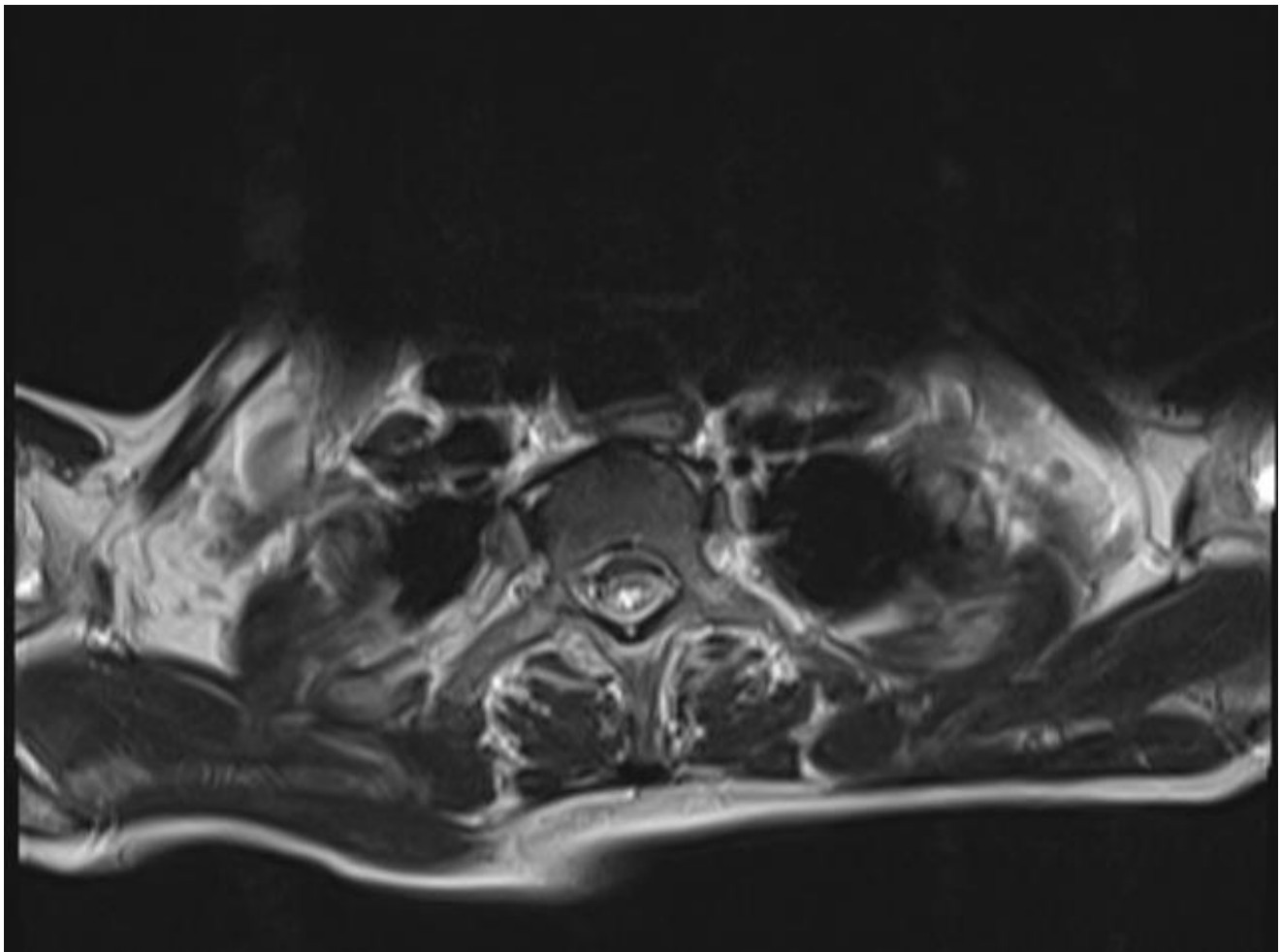

**Figure 3.** T2-weighted image of the spinal cord: hyperintense lesions in the white matter and in the posterior portion of the spinal cord.

## 4. Discussion

In this paper, we report a case of longitudinally extensive transverse myelitis due to primary CMV infection in an immunocompetent host. LETM is a rare entity with severe and irreversible clinical consequences if it is not treated promptly; therefore, it is also necessary to identify the etiological agent to prevent future reactivations in the spinal cord. Patients with TM present with rapid-onset weakness, sensory disturbances, and bladder dysfunction [1]. The types of TM according to their causative agent are: idiopathic TM and secondary TM to some diseases [11]. TMs have diverse etiologies including infection (West Nile virus, human immunodeficiency virus (HIV), human T-cell lymphotrophic virus (HTLV)-1, Borrelia, Mycoplasma, and Treponema pallidum), vaccination, and other diseases. Demyelinating, systemic, and vascular diseases have also been observed (e.g., multiple sclerosis, neuromyelitis optica (NMO), Sjögren's syndrome, sarcoidosis, paraneoplastic syndromes, systemic lupus erythematosus (SLE), and spinal cord infarction) [5].

Following a thorough PubMed search (1940–2014) for CMV-associated TM, only 12 cases were identified, and 4 cases were excluded from the review, as they did not fulfil the 'well-documented' criteria [5]. CMV myelopathy may be due to direct cytotoxic viral damage to myelin and may be a consequence of the overexpression of the immune-mediated humoral immune system due to the similarity of the CMV capsid protein to the oligodendrocyte protein MOG34–56 [1,12]. These two stages can be consecutive or simultaneous. In our case, it seems that these two mechanisms were present due to the

favorable clinical response to immunosuppressive and antiviral treatment. Our patient presented elevated serological IgM and IgG antibodies. In addition, this agrees with the fact that in a similar case report, it was demonstrated that the IgM titer was elevated in all patients, except for two, as did the patient in our case [5]. The presence of IgM and IgG antibodies could be related to the reactivation of a previous CMV infection or reinfection with a different CMV strain due to the detection of CMV on the FilmArray panel in this case.

MRI has an important role in the diagnosis and classification of TM, as it allows us to identify hyperintense spinal cord injury in T2-weighted images, and when it affects at least three vertebral segments in the sagittal slices, TM is classified as LETM [3]. T2- weighted MRI demonstrated an intramedullary increased signal in the central portion of the spinal cord as the most common finding [5]. MRI images in immunocompetent patients with TM show that the cervicothoracic cord is affected more frequently, as in our case, unlike immunosuppressed patients, in which tropism has been seen in the cauda equina or the conus medullaris [13]. The aforementioned findings coincide with the RMI in the case report presented, in which hyperintense cervicothoracic spinal cord involvement was observed in the T2-weighted image. This was specifically seen in patients with NMO (anti-AQP4 Ab) associated with patients with SLE and Sjögren's disease [3,14]; while there is evidence of markedly hyperintense foci on T2 described as "Bright Spotty Lesions", characteristic for NMO, in this case, anti-AQP4 antibody sera were not performed. However, the results of the IgM serological test for cytomegalovirus was positive, which confirmed the acute installation of the infectious image.

The mean values of the CSF biochemical test results performed in the review of nine well-documented cases of CMV-associated TM were as follows: CSF protein, $229.75 \pm 157$ mg/dL (range 37–480 mg/dL); CSF leucocyte count, $105 \pm 135.4$ cells/mm$^3$ (range 0–350 cells/mm$^3$). The majority of cases showed CSF pleocytosis with lymphocyte predominance [5]. With regard to this fact, in this case, CSF remained in the average ranges previously considered. CSF that demonstrated mononuclear pleocytosis may suggest that some viral agents are the etiology of TM [11], and due to the multiple involvements in MRI, the case was classified as LETM. The BioFire FilmArray® Meningitis/Encephalitis Panel (FAMEP) is designed to rapidly and accurately detect common multiple pathogens that cause central nervous system (CNS) infection, including viruses, bacteria, and yeast [15]. However, it can be negative when LETM is in post-infectious stage. In our case, the identification of CMV using this method demonstrated the acute viral process.

Regarding treatment, corticosteroids were used as pulses of methylprednisolone due to their immunosuppressive and anti-proliferative activity, in addition to the fact that they reduced the inflammatory response [16]. Ganciclovir was also administered with a good clinical response and without severe adverse effects. There are similar reports that explain that the use of ganciclovir does not alter the clinical improvement after the use of corticosteroids [12]. Apart from corticosteroids, several acute therapies have been used in patients with transverse myelitis including plasma exchange, intravenous immunoglobulin (IVIg) and cyclophosphamide. Limited evidence exists to support IVIg or cyclophosphamide, but there is some evidence to support the use of plasma exchange therapy. In one retrospective review, transverse myelitis treated with corticosteroids plus plasma exchange had twice as much improvement when compared with patients treated with corticosteroids alone [17].

In conclusion, CMV infection should be considered as a cause of LETM, especially in immunocompetent hosts, and early therapy with ganciclovir and methylprednisolone may improve prognoses. Magnetic resonance imaging is an important key to diagnosis and can exclude another possible etiologies. In addition, timely treatment should be based on serological and molecular studies. Additionally, is very important to mention that CMV disease must be excluded via CSF and blood analyses before starting any therapy.

**Author Contributions:** Conceptualization, R.M. and M.-F.S.V.; methodology and investigation, R.M. and M.-F.S.V.; writing—original draft preparation R.M. and M.-F.S.V.; writing—review and editing, R.M. and M.-F.S.V. All authors have read and agreed to the published version of the manuscript.

**Funding:** This research received no external funding.

**Institutional Review Board Statement:** Ethical review and approval were waived for this study, due to the fact that only data and images produced for the management of the patient and collected after her death were used.

**Informed Consent Statement:** Patient consent was waived due to the fact that only data and images produced for the management of the patient and collected after her death were used.

**Data Availability Statement:** Not applicable.

**Conflicts of Interest:** The authors declare no conflict of interest.

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
