# Peer review of "Longitudinally Extensive Transverse Myelitis Associated with Cytomegalovirus Infection in an Immunocompetent Patient"

_2036-7481, doi:10.3390/microbiolres13030036_

Round 1

Reviewer 1 Report

Overall, a case of some interest. Would suggest the following:

1. Please include result of anti-MOG testing; I only see anti-aquaporin-4 noted. Additionally, from infectious perspective, please include testing for West Nile virus/arboviruses, Lyme disease, and also explicitly state that more common viral PCRs in CSF (e.g. HSV-1 and 2, VZV, enterovirus) were negative.

2. Were other features of CMV infectious present (e.g. lymphadenopathy, hepatitis/elevated liver enzymes)? Would be important to include. 

3. Was CMV detected in blood in addition to CSF?

4. There is confusion of the acronyms. Tranverse myelitis is appropriately abbreviated TM, but then referred to as MT and longitudinally extensive tranverse myelitis is incorrectly abbreviated as MTLE rather than LETM. Please correct throughout.

5. The authors state that "so the efficacy of the antiviral agent is high in the first days after the onset of the clinical picture and uncertain when the time of illness is prolonged due to because MTLE is also the result of an immune-mediated mechanism". This sentence is difficult to follow grammatically and the evidence to support high efficacy of antiviral therapy in the first days of immunocompetent CMV LETM is lacking; would remove this statement

6. Please change "and early therapy with ganciclovir and methylprednisolone improves prognosis" to "and early therapy with ganciclovir and methylprednisolone may improve prognosis"; cannot make definitive statements from one case

7. There are grammatic/syntactic errors throughout, would benefit from proofreading

Author Response

RESPONSE TO REVIEWER 

  1. Please include result of anti-MOG testing; I only see anti-aquaporin-4 noted. Additionally, from infectious perspective, please include testing for West Nile virus/arboviruses, Lyme disease, and also explicitly state that more common viral PCRs in CSF (e.g. HSV-1 and 2, VZV, enterovirus) were negative.

Response 1: anti-MOG and anti-aquaporin-4 testing were not performed, this information has been corrected. Also Testing for West Nile virus/arboviruses, were not included , because of the absence of relevant data such as travel or residence in endemic areas. Also Lyme test was not performed due to the low prevalence of the disease in Perú .

  1. Were other features of CMV infectious present (e.g. lymphadenopathy, hepatitis/elevated liver enzymes)? Would be important to include. 

Response 2: Patient did not present lymphadenopathy , elevated liver enzymes, or another relevant data.

  1. Was CMV detected in blood in addition to CSF?

Response 3 :Serological CMV test in blood was not performed.

  1. There is confusion of the acronyms. Tranverse myelitis is appropriately abbreviated TM, but then referred to as MT and longitudinally extensive tranverse myelitis is incorrectly abbreviated as MTLE rather than LETM. Please correct throughout.

Response 4: The correction has been done ,  longitudinally extensive myelitis (LETM), Tranverse Myelitis (TM).

  1. The authors state that "so the efficacy of the antiviral agent is high in the first days after the onset of the clinical picture and uncertain when the time of illness is prolonged due to because MTLE is also the result of an immune-mediated mechanism". This sentence is difficult to follow grammatically and the evidence to support high efficacy of antiviral therapy in the first days of immunocompetent CMV LETM is lacking; would remove this statement

Response 5: The statement was already removed , because of the grammatical difficulty and lack evidence.

  1. Please change "and early therapy with ganciclovir and methylprednisolone improves prognosis" to "and early therapy with ganciclovir and methylprednisolone may improve prognosis"; cannot make definitive statements from one case

Response 6: This sentence was changed to “and early therapy with ganciclovir and methylprednisolone may improve prognosis” .May provides us with different options, and gives us a better context on the treatment of this particular case.

  1. There are grammatic/syntactic errors throughout, would benefit from proofreading

Response 7: paper was reviewed , correcting grammatical mistakes in the wording.

Thank you for your comments, without further ado, I look forward to your reply.

Reviewer 2 Report

Thank you for the opportunity to review this interesting manuscript about  CMV Longitudinal transverse myelitis. However, the authors need  to thoroughly revise the grammar of this manuscript. Please consider the following comments:

ABSTRACT

- Consider abbreviating Magnetic resonance imaging as (MRI) is a recurring term in the text.

- In the abstract the term Longitudinally extensive transverse myelitis is used and is abbreviated as (LTM) and this abbreviation is not used throughout the text

INTRODUCTION

-The manuscript requires editing.

- The abbreviation MTLE seems inappropriate to me and seems to be an abbreviation of the Spanish "Mielitis Transversa Longitudinalmente Extensa" The use of this abbreviation makes it difficult to understand the text. I recommend that the abbreviation MTLE be replaced by LTM.

- I recommend that in the introduction and discussion you cite and discuss the article by Ferhat Arslan DOI: 10.3109/00365548.2014.964763

CASE PRESENTATION

- It is incorrect to report the section title as material and methods and the section title should be changed to case report. I suggest reviewing other case reports in the journal to understand the format requested.

-line 37 page 1 suggest to note the ethnicity of the patient and the region where the patient was seen.

-line 40 page 1 I consider that the term bladder ballon is not a correct medical term I suggest describing the patient as having acute urinary retention.

- Before the neurological examination, indicate if there were any abnormal findings in the neck, thorax and abdomen.

- line 51 page 2 I consider that it is not necessary to include a results section since the description of the clinical case should be included.

- Consider shortening the lab results section. I would recommend rewriting as HTLV, HIV1-2, hepatitis B, hepatitis C, herpes simplex virus (HSV)-1-2 and Epstein Barr were all negative.

- line 55 page 2 Please specify what type of syphilis test was used.

- I suggest that only abnormal findings be described in the interpretation of cerebrospinal fluid.

Discussion

- The use of different abbreviations makes reading difficult. I recommend unifying abbreviations throughout the text.

- line 118 page 6 the authors wrote one word "disease9". Please clarify if it was a mistake in trying to quote [9].

- line 59 page 3 the authors wrote "double-stranded DNA, cytoplasmic, perinuclear, antiphospholipid antibodies, and anti-aquaporin-4. they were negative."  In line 129 page 6 the authors wrote  "anti-AQP4 antibody studies were not performed", please clarify if this test was performed.

- I understand that the patient was not tested for Lyme. Please clarify if in your region there is information on the epidemiology of this disease since neuroborreliosis may be a cause of transverse myelitis.

  Consider abbreviating Magnetic resonance imaging as (MRI)

Author Response

RESPONSE REVIEWER 

1.-ABSTRACT

- Consider abbreviating Magnetic resonance imaging as (MRI) is a recurring term in the text.

- In the abstract the term Longitudinally extensive transverse myelitis is used and is abbreviated as (LTM) and this abbreviation is not used throughout the text

Response 1: The abbreviation for Magnetic Resonance Imaging was modified to MRI. Also for Longitudinally Extensive Transverse Myelitis was changed to LETM.

2.-INTRODUCTION

-The manuscript requires editing.

- The abbreviation MTLE seems inappropriate to me and seems to be an abbreviation of the Spanish "Mielitis Transversa Longitudinalmente Extensa" The use of this abbreviation makes it difficult to understand the text. I recommend that the abbreviation MTLE be replaced by LTM.

- I recommend that in the introduction and discussion you cite and discuss the article by Ferhat Arslan DOI: 10.3109/00365548.2014.964763

Response 2: The manuscript was reviewed for editing . Term MTLE was replaced for LTM and we introduce the suggested article in the discussion and it is already cite.

3.-CASE PRESENTATION

- It is incorrect to report the section title as material and methods and the section title should be changed to case report. I suggest reviewing other case reports in the journal to understand the format requested.

-line 37 page 1 suggest to note the ethnicity of the patient and the region where the patient was seen.

-line 40 page 1 I consider that the term bladder ballon is not a correct medical term I suggest describing the patient as having acute urinary retention.

- Before the neurological examination, indicate if there were any abnormal findings in the neck, thorax and abdomen.

 -line 51 page 2 I consider that it is not necessary to include a results section since the description of the clinical case should be included.

- Consider shortening the lab results section. I would recommend rewriting as HTLV, HIV1-2, hepatitis B, hepatitis C, herpes simplex virus (HSV)-1-2 and Epstein Barr were all negative.

- line 55 page 2 Please specify what type of syphilis test was used.

- I suggest that only abnormal findings be described in the interpretation of cerebrospinal fluid.

Response 3: We reviewed other case reports examples of the journal and the point was modified. In line 37 is included the patient’s ethnicity. Line 40 term was changed to acute urinary retention . In neurological examination there were any abnormal findings in neck , thorax and abdomen.

Line 51 we include de results , and this statement was modified to: HTLV, HIV1-2, hepatitis B, hepatitis C, herpes simplex virus (HSV)-1-2 and Epstein Barr were all negative.

line 55 page 2 includes the syphilis type of test , it was a fast test. In CFS we include only the abnormal findings .

 4.-Discussion

- The use of different abbreviations makes reading difficult. I recommend unifying abbreviations throughout the text.

- line 118 page 6 the authors wrote one word "disease9". Please clarify if it was a mistake in trying to quote [9].

- line 59 page 3 the authors wrote "double-stranded DNA, cytoplasmic, perinuclear, antiphospholipid antibodies, and anti-aquaporin-4. they were negative."  In line 129 page 6 the authors wrote  "anti-AQP4 antibody studies were not performed", please clarify if this test was performed.

- I understand that the patient was not tested for Lyme. Please clarify if in your region there is information on the epidemiology of this disease since neuroborreliosis may be a cause of transverse myelitis.

Response 4: The abbreviations were unified . Line 118 the term “disease9” was replaced to “disease [9]. Line 59 the data was clarified and modified to: Anti-AQP4 antibody test was not performed. And The epidemiology of the region where the patient lives was clarified.
